# PARAMETER-EFFICIENT INVARIANCE VIA EQUIVARIANT OBJECT DETECTION NETWORKS

## ABSTRACT

Leading convolutional object detection networks like the YOLO series have proven highly effective in generalizing across feature symmetries such as rotation and scale, but often do so at the cost of increased parameter counts that complicate model training and inference. Following recent work in the field of equivariant convolutions, this work seeks to exploit rotational and scale symmetries present in image data for parameter-efficient invariance. We propose three object detection models based on the YOLOv9-t (Y9t) architecture that incorporate equivariant convolutions for rotation $C_4$-Y9t, scale, S-Y9t or both $C_4$S-Y9t. To directly evaluate rotation- and scale-invariant methods in object detection networks, we propose the novel MNIST Object-Detection (MOD) datasets. By incorporating equivariant methods into the Y9t network while maintaining a similar number of trainable parameters, mean average precision (mAP) is additively increased by 16.8%, 15.2%, and 3% on unseen rotation, scale, and rotation-scale transformed partitions of the MOD datasets, respectively. Further, the proposed models generalize to other object detection datasets (COCO 2017), with our best model increasing mAP by 4.9%.

## 1 INTRODUCTION

When designing a convolutional object-detector for two-dimensional images, one must consider the tradeoffs between the size of the model (typically measured in the number of trainable parameters), the inference time of the model (measured in milliseconds on a given processing unit or GFLOPs) and the overall output quality of the model (measured by mean Average Precision, (mAP) a metric combining classification and bounding box errors). The latter two metrics are generally related to the size of the model, i.e. large models tend to produce higher inference times and mAP scores than small ones.

The goal of this research is to study the impact on invariance from exploiting natural symmetries that occur in images and convolutional neural networks to improve quality while maintaining the approximate number of parameters. In particular, we explore the use of rotation and scale equivariant convolutions in the YOLOv9-t (Y9t) object detection network (Wang et al., 2024; Jocher et al., 2023). Equivariance[1] is an idea closely related to invariance. A method is said to be *invariant* to given transformations if, regardless of those transformations being applied to the input data, the output remains unchanged. A method is said to be *equivariant* to given transformations if, when those transformations are applied to the input, the output is transformed similarly. We focus on utilizing symmetry groups, such as rotation or scale groups, to compute equivariant convolutions in object detectors. These equivariant convolutions are discussed in Section 2.

Previous work has been conducted on equivariance in object detection. We briefly discuss this work and some datasets used to evaluate it in Section 3. Section 4.1 proposes three models based on the Y9t architecture with the goal of improving the size-quality trade-off: $C_4$-Y9t, S-Y9t, and $C_4$S-Y9t. We aim to demonstrate a concrete increase in transformation robustness resulting from the inclusion of equivariant convolutions, while simultaneously maintaining the overall number of learnable parameters.

---

[1]A more detailed explanation of equivariance can be found in Appendix A.2.

To measure the robustness of our methods to specific transformations, in Section 4.2 we propose the MNIST Object-Detection (MOD) datasets. Briefly, our MOD Datasets Generation Framework[2] constructs a MOD dataset by placing transformed MNIST digits on a uniform background. To simulate natural image conditions where objects occur in complex scenes, a "PSbg" variant is proposed that adds a background that is matched in spectral content to the standard variant (filtered noise). The proposed datasets allow us to analyze the difference in mAP between the Y9t model and our proposed models by training on a dataset with minimally transformed digits and testing on versions constructed using transformed digits. By training and testing in this manner, we are able to gauge any improvements in the handling of features transformed by specific symmetry groups. We then generalize these results by training all models on the COCO 2017 dataset from Lin et al. (2014) following hyperparameters used with YOLOv9. Experimental results are given in Section 5.

This work makes three contributions. We implement several techniques for equivariant convolutions in the Y9t architecture, with the aim of explicitly analyzing how the use of equivariance affects invariance to various transformations in a similarly sized model. Secondly, we propose the MNIST Object Detection (MOD) datasets to allow researchers concretely and consistently gauge changes to robustness in the face of transformations. Finally, we show that our proposed models are effective on other general-purpose datasets where object scales vary within a scene, but orientation is generally consistent, i.e. COCO 2017 (Lin et al., 2014).

## 2 EQUIVARIANT CONVOLUTIONS

A great deal of research has been conducted regarding rotational (Cesa et al., 2022; Cheng et al., 2019) and scale (Sosnovik et al., 2021a; Tiezzi et al., 2023) equivariant convolutions in recent years. We focus on the approach using the cardinal rotation group $C_4$ presented by Cohen & Welling (2016) for rotation equivariance and Worrall & Welling (2019)'s kernel dilation approach using the scaling semigroup S for scale equivariance. We select these methods due to our preliminary testing showing computational efficiency and high mAP impact in the Y9t model.

Though we focus on the $C_4$ group, we note other approaches for rotationally equivariant convolutions. Cohen & Welling (2016) also suggest augmenting this rotation group with reflections, denoted $D_4$, by using a translation-supporting variant $p4m$. Other methods for rotational equivariance use the continuous $SO(2)$ group, allowing for non-cardinal rotations using interpolation (Weiler & Cesa, 2019; Cesa et al., 2022).

In addition to the usage of rotational symmetries, we explore the exploitation of scale symmetries. Ideally, a single kernel should be scale invariant, i.e., it should detect all instances of some feature regardless of its relative size within an image. Two popular methods for implementing scale equivariance in convolutional neural networks are decomposed kernels (Gao et al., 2022; Sosnovik et al., 2020; 2021a;b; Zhu et al., 2022), and dilated kernels (Worrall & Welling, 2019; Tiezzi et al., 2023; Jansson & Lindeberg, 2020). We focus on the dilation approach of Worrall & Welling (2019) using the scale semigroup *S*. We choose this kernel dilation approach because while limited to integer scales, it is much more time-efficient than decomposed kernels due to explicit kernel dilation support in machine learning libraries.

### 2.1 ROTATIONAL EQUIVARIANCE

A method for computing the convolutional integral that takes advantage of the symmetrical properties of images via groups is provided by Cohen & Welling (2016). Here, rotational equivariance is achieved through the cardinal rotation group $C_4$, built in practice using the $p4$ group (which composes all translations and rotations about an arbitrary origin in a square grid).

Given data from $\mathbb{Z}^2$, the lifting layer for the $C_4$ group applies a single kernel in each of the four orientations supported by the symmetry group, yielding four distinct streams of data in a dimension representing orientation. To maintain this equivariance throughout subsequent layers while allowing all orientation channels to interact, additional kernels are added. Specifically, $C_4$ requires three additional kernels, one for each additional rotation. These kernels are iteratively transformed and applied to the relevant data stream to maintain four orientation channels.

---

[2]To be published on GitHub alongside this paper and included in supplementary material.

Equipped with using the $C_4$ group, Cohen & Welling (2016) derive their convolutional method as follows. To compute a group equivariant convolution, first consider the plane symmetry group $G$. $G$ is called *split* if any transformation $g \in G$ can be decomposed into a translation $t \in \mathbb{Z}^2$ and a transformation $s$ such that $s$ leaves the origin invariant. For the $p4$ group, we have $g = ts$, where $t$ is a translation from the origin and $s$ a rotation about the origin. Using the split of $G$, and the fact that the concrete instantiations of transformations $g, h \in G$ are $L_g, L_h$ respecting $L_g L_h = L_{gh}$, we have a convolution of form

$$f \star \Psi(ts) = \sum_{h \in X} \sum_k f_k(h) L_t [L_s \Psi_k(h)]. \tag{1}$$

where $X = \mathbb{Z}^2$ in layer one, $X = G$ in subsequent layers, $\Psi$ is the convolutional filter, and both $L_t$ and $L_s$ are concrete instantiations of a translation $t$ and some arbitrary transformation $s$ that leaves the origin invariant, respectively (Cohen & Welling, 2016). This equation iterates through all equivariant data streams, applying each transformation of kernel $k$ to each image patch $f_k(h)$. For $\mathbb{Z}^2$, only one data stream exists, but for $C_4$ there are four distinct streams, one for each orientation.

## 2.2 Scale Equivariance

Worrall & Welling (2019) achieve scale equivariance by first using a lifting layer. The input data is lifted to a scale space by low-pass filtering via Gaussian blur kernels to represent different scales of the data while maintaining the resolution. For convolutions, the same kernel is run over the data multiple times; different dilations and input scale representations are used for each pass to process the data at multiple scales. As with rotation, an additional dimension is used to contain the data pertaining to each particular scale. When used as an equivariant method, a dilated kernel allows for interactions between scales, and while in many cases this interaction not beneficial (Zhu et al., 2022), a small amount has been useful in practice. Using inter-scale interaction on every other convolution balances boundary effects stemming from the discretization of S and network expressiveness (Worrall & Welling, 2019). Additional parameters are required to allow for mixing information between scales.

Worrall & Welling (2019) derive the dilated kernel convolution by first noting that the scale-space correlation $\star_S$ is restricted to a discretization of the scale-space where dilation parameter $A$ is limited to $A_k = 2^{-k} I$ for $k \geq 0$ and zero-scale $\sum_0 = \frac{1}{4} I$. Further, denoting $f(2^{-k} I, \cdot)$ as $f_k(\cdot)$, the formula for a correlation on the lifted signal is given in Equation 2:

$$[\Psi \star_S f]_k(z) = \sum_{\ell \geq 0} \sum_{y \in Z^d} \Psi_\ell(y) f_{\ell+k}(2^k y + z). \tag{2}$$

## 3 Related Work

Implementing equivariance in object detectors for 2D images has been explored previously with respect to rotation or scale (Han et al., 2021; Yan et al., 2023; Wu et al., 2023; Lee et al., 2024; Sosnovik et al., 2021b; Maurel et al., 2023; Cho et al., 2024). In Section 3.1, we review rotation- and scale-equivariant models. Each equivariant object detector typically implements a single equivariant method throughout the model for application to a dataset with a high degree of feature symmetry with respect to the chosen equivariant method. Given the relevant of datasets, we then discuss datasets that have been previously used to evaluate robustness to transformations, invariance, or equivariance in convolutional neural networks in Section 3.2.

### 3.1 Equivariant Object Detection

The most relevant models to this work are the EQ-YOLO models introduced by Maurel et al. (2023) because they similarly use equivariance as a powerful form of invariance. These models make use of the $C_8$ group, a subset of the $SO(2)$ group with eight equidistant rotations (Cesa et al., 2022), by replacing many blocks in the YOLOv8 model (Maurel et al., 2023; Jocher et al., 2023) with their equivariant counterparts. A separate projection layer is used for the classification output and the bounding box output. Importantly, while this approach reduced the number of learned

model parameters by a factor of three or more, inference was computationally inefficient. The EQ-YOLO models showed improved mAP on both their OD-MNIST dataset as well as on aerial imagery (Maurel et al., 2023). The experimental setup for the modified MNIST dataset only tests rotational equivariance, allows for rotation and mirroring as data augmentations rather than preserving the transformations as unseen data, and reduces the number of parameters rather than comparing a similarly sized model (Maurel et al., 2023). As such, the experiment does little to explicitly show the invariance gained from the equivariance method.

ReDet (Han et al., 2021) provides rotational equivariance in oriented object detectors for aerial imagery by using an equivariant backbone, with experiments on $C_4$, $C_8$, and $C_{16}$. An interpolation method is applied on the orientation channels for intermediate orientations. In addition to the rotation of the bounding box, rotational equivariance is used to provide invariance when determining the classification of features. This is further improved by altering the structure to a rotation-equivariant, bidirectional feature fusion feature pyramid network (Yan et al., 2023). Similarly, a two-way feature fusion network, a novel loss function based on Gaussian Wasserstein distance, and a more scale invariant backbone are used by Wu et al. (2023). Each modification improved mean average precision on aerial object detection, with the base model and the first improvement showing a large parameter reduction. Another rotation equivariant object detector on $C_8$, FRED (Fully Rotation-Equivariant Oriented Object Detector) uses a vector representation for bounding boxes along with rotationally equivariant deformable convolutions for a higher robustness achieving several state of the art results on aerial object detection datasets (Lee et al., 2024).

Sosnovik et al. (2021b) explore the idea of scale equivariance for Siamese tracking, a version of object tracking where two input images are compared, using decomposed convolutions for scale equivariance to improve precision. For object detection, Cho et al. (2024) use a steerable scale equivariant backbone network for an autonomous driving problem. To reduce the error in high-frequency details due to the equivariant method, they dedicate distinct convolutions to high- and low-frequency features. Their approach leads to a reduction of the overall equivariance error and shows improved precision on autonomous driving datasets (Cho et al., 2024).

### 3.2 Datasets Measuring the Efficacy of Equivariance

To systematically evaluate how equivariant methods affect object recognition performance, we require datasets in which object orientation and scale are systematically varied. Here, we present several such datasets based on the MNIST hand-written digit dataset (LeCun et al., 2010) consisting of 10 basic features. In particular, we discuss MNIST-rot (Larochelle et al., 2007), and MNIST-scale (Sohn & Lee, 2012), and for object detection in particular, OD-MNIST (Maurel et al., 2023), and T-MNIST and S-MNIST (Sosnovik et al., 2021b).

The MNIST-rot and MNIST-rot-back-image datasets (Larochelle et al., 2007) are designed for classification tasks. Each $28 \times 28$ pixel image in the datasets contains a single digit rotated by a uniformly random angle from 0 to $2\pi$ radians. MNIST-rot-back-image, seen in Figure 1a, places a random patch from a black and white image (selected from a set of 20 internet images) in the background of each digit image. Adding these patches imposes a sort of noise on the image to which a model must become robust. Similarly, Sohn & Lee (2012) provide a classification dataset that scales the digits of MNIST. MNIST-Scale maintains the image patch size of $28 \times 28$, and scales the digits inside the image by a uniformly random amount in the range $[0.3, 1]$, as seen in the four samples in Figure 1b. This scaled dataset, like MNIST-rot, has a variant with uniform noise in the range $[0, 1]$ as a background, which achieves a similar effect to random patches.

Building on the MNIST-rot and MNIST-Scale, Sosnovik et al. (2021b) introduce the T-MNIST and S-MNIST datasets for testing scale equivariance in object tracking models. T-MNIST consists of up to 8 MNIST digits with backgrounds from an object tracking dataset, with random translations induced on each digit (Sosnovik et al., 2021b). S-MNIST, extends T-MNIST by applying a smooth scaling in the range $[0.67, 1.5]$ to the digits, as shown in Figure 1c.

Maurel et al. (2023) provide OD-MNIST, an object detection dataset for evaluating equivariance methods. The dataset consists of $134 \times 134$ images with randomly positioned MNIST digits. They create a training set with non-rotated digits and a testing set with rotated digits, which can be used to assess the impact of equivariant methods in object detectors. OD-MNIST is a large step forward for

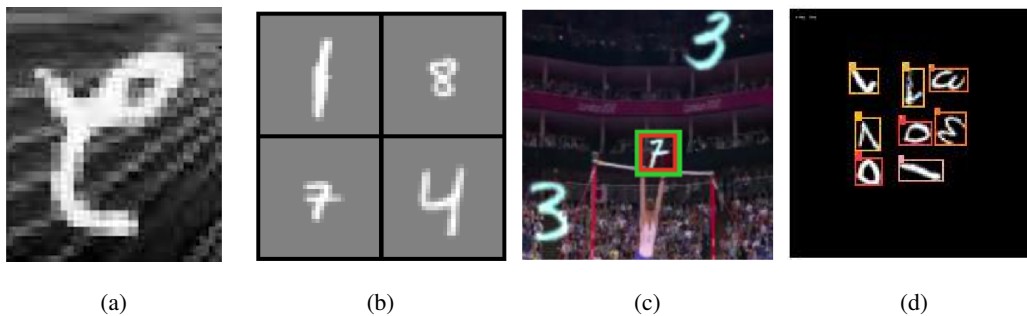

(a)          (b)          (c)          (d)

Figure 1: Scaled example images from existing transformed MNIST datasets. (a) MNIST-rot-back-image (Larochelle et al., 2007). (b) MNIST-Scale (Sohn & Lee, 2012). (c) S-MNIST (Sosnovik et al., 2021b). (d) OD-MNIST's testing set (Maurel et al., 2023).

object detection, but fails to add any background noise to the image. An example from their testing set, showing various rotations can be seen in Figure 1d.

## 4    METHODS

This section first explains our proposed variants of the Y9t model. We then discuss their construction, limitations, and equivariant hyperparameters. Finally, we describe our proposed MOD datasets, and their Power Spectrum background (or "-PSbg"), variants.

### 4.1    MODELS

We propose three models to evaluate the effectiveness of equivariant convolutions as a form of invariance on object detection tasks: $C_4$-Y9t, S-Y9t, and $C_4$S-Y9t. These models are constructed using group equivariant convolution on the $C_4$ group (Cohen & Welling, 2016), and scale equivariant convolution on the semigroup S (Worrall & Welling, 2019). Each model replaces convolutions with equivariant counterparts (see Table 1). These models are designed to systematically assess and quantify invariance to several common transformations while maintaining a similar number of parameters. We implement equivariant versions of rep-style blocks, up-sampling layers, CSPBottleneck, RepBottleneck, Bottleneck, RepCSP, RepNCSPELAN4, ELAN1, AConv, ADown, and SP-PELAN using the Ultralytics implementations of these libraries as non-equivariant baselines (Jocher et al., 2023).

Table 1: Construction, parameters, and inference time of proposed models

| Model | Equivariance (Backbone) | Equivariance (Neck) | Learnable Parameters | Inference Time (ms, NVIDIA v100) |
|---|---|---|---|---|
| Y9t (Wang et al., 2024; Jocher et al., 2023) | N/a | N/a | 2 128 720 | 2.4 |
| $C_4$-Y9t | $C_4$ | $C_4$ | 1 939 888 | 5.5 |
| S-Y9t | S | S | 2 086 389 | 34.0 |
| $C_4$S-Y9t | $C_4$ | S | 2 128 301 | 21.9 |

Each model is based on the core Y9t architecture (Wang et al., 2024), the YOLOv9 variant with the fewest learned parameters. This particular version was selected for two reasons. First, smaller models were expected to benefit more from transformation invariance because their parameters are expected to be nearly saturated with essential information for object detection and classification, such as shapes, patterns, and curves. Second, because most equivariant methods have been found to be computationally expensive, we opted for the smallest model to minimize the time required for training and inference.

Although equivariant methods can leverage existing parameters, some implementations may introduce additional parameters beyond those needed for standard convolutions to enhance computational efficiency or facilitate interaction between transformations. To avoid increasing the overall size of the model, we constrained each block in the model, and by extension, each equivariant convolutional

layer, to have fewer channels than the corresponding amount in Y9t. $C_4$-Y9t has 50% of the original channels, S-Y9t has 84% of the original channels, and $C_4$S-Y9t has 75% and 50% of channels in the neck and backbone, respectively. No reduction was applied to the first layer of the model to avoid an information bottleneck. These percentages were chosen arbitrarily such that each variant approximates the number of parameters in Y9t. Detailed information about the number of parameters in each model, equivariant substitutions, and inference time is given in Table 1 (see Appendix A.1 for explicit construction details).

By using these equivariant convolutions, our models must include lifting layers and projections back to $\mathbb{R}^2$ space. For $C_4$-Y9t, the additions are straightforward, with a lifting layer replacing the first convolution of Y9t, and a maximal projection inserted before the detection heads. Maximal projection was chosen since it appeared to have the most substantial impact on model performance during preliminary testing. In S-Y9t, contrary to that reported by (Sosnovik et al., 2020), we found projection prior to up- or down-sampling harmed output quality and thus these projections were omitted. We implement an average projection for the S scale semi-group, in contrast to other scale equivariance works, for similar rationale. When concatenating with the skip connections to the neck of the model, both $C_4$-Y9t and S-Y9t concatenate channel dimensions.

$C_4$S-Y9t is the most complex model we consider. Here, we lift to the $C_4$ group at the first convolutional layer, and maintain this group throughout the backbone of the model. Unlike S-Y9t, we find that projecting prior to resampling layers improves quality and thus include the practice in $C_4$S-Y9t. Since every skip connection to the neck of the model coincides with a scaling operation, the data is projected then concatenated in $\mathbb{R}^2$ before being lifted to S. Following the spatial pyramid pooling in $C_4$, we project and lift to S for the neck of the model.

In both proposed models using scale equivariance, we follow the implementation by Worrall & Welling (2019), using four scales in powers of $\frac{1}{2}$ with an assumed input scale of 0.5. Similarly, we allow for a small amount of inter-scale interaction on every second convolution in the model. Finally, we use the $1 \times 1$ kernel scale equivariant convolution proposed by Sosnovik et al. (2020) in the construction of any blocks that require such a kernel because it is fast and effective in our context.

## 4.2 MNIST Object Detection Datasets

Many of the experiments on equivariance in object detection models are measured on datasets which consist of features with a high degree of symmetry relevant to the equivariant method. For example, rotation equivariant object detectors are typically tested on aerial imagery, a domain with features in many orientations. Similarly for scale, the autonomous driving dataset used by Cho et al. (2024) consists of features in many scales. An important question is how to explicitly evaluate the effect of the equivariant method on the robustness of the network in regards to a particular symmetry group.

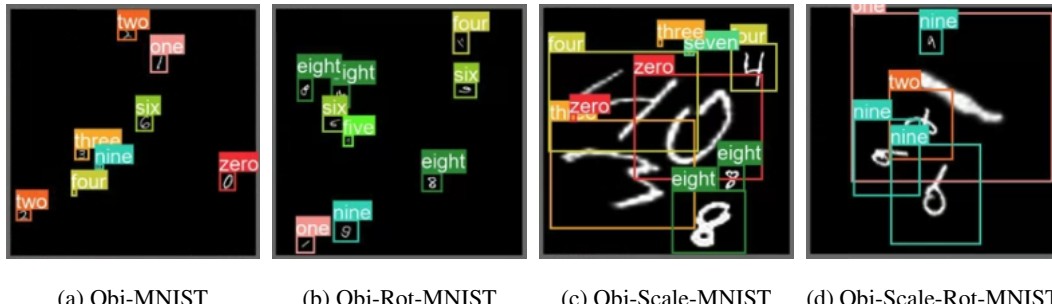

|     (a) Obj-MNIST     |     (b) Obj-Rot-MNIST     |     (c) Obj-Scale-MNIST     |     (d) Obj-Scale-Rot-MNIST     |

Figure 2: Examples from our proposed Obj-MNIST datasets.

We propose the MNIST Object Detection (MOD) datasets, exemplified in Figure 2. These datasets are built for object detection using the digits from the MNIST dataset (LeCun et al., 2010) and consist of four configurations: *Obj-MNIST*, a dataset with minimal transformations; *Obj-Rot-MNIST*, a dataset with rotation in the range of $[0, 359]°$ applied to the digits; *Obj-Scale-MNIST* a dataset with scaling in the range of $[0.5, 16]\times$ applied to the digits; and *Obj-Scale-Rot-MNIST*, a dataset with both prior transformations applied to digits (see Table 2). Each dataset consists of 10,000 training

samples, 2,000 testing samples and 2,000 validation samples. No unique digits overlap between these three splits.

Table 2: Transformations in proposed individual Obj-MNIST datasets

| Dataset | Obj-MNIST | Obj-Rot-MNIST | Obj-Scale-MNIST | Obj-Scale-Rot-MNIST |
|---|---|---|---|---|
| **Rotation** | N/a | $[0, 359]°$, $1°$ increments | N/a | $[0, 359]°$, $1°$ increments |
| **Scale** | $[0.5, 2.0]\times$ | $[0.5, 2.0]\times$ | $[0.5, 16.0]\times$ | $[0.5, 16.0]\times$ |

For each image of these datasets, a random number (from five to fifteen) of handwritten digits are selected from the MNIST dataset (LeCun et al., 2010), transformed, and randomly placed in a blank $640\times640$ pixel image. Masks are used to cut digits from the original blank $28\times28$ pixel background, and logical operators ensure that digits do not overlap. Digits are then transformed by each specified transformation in sequence, and a position is randomly chosen. Logical masks are used to ensure no digits directly overlap. We allow for up to 10 random placement attempts before giving up and moving on to the following digit. Any of the bounding boxes may overlap, allowing for digits to be much closer to one another than with prior approaches. A further benefit is the potential for numbers nested inside each other without overlap. For example, in the scaling dataset, a small "1" may be placed in the center of a large "0". Further, bilinear interpolation is employed when rotating or scaling to prevent aliasing in digits. The framework to create these datasets is available on GitHub[3].

Following Larochelle et al. (2007), we further propose the Power Spectrum background (or "-PSbg"), variants of the four datasets. Here, we calculate the average power spectrum across images of each dataset and use phase scrambling to produce filtered noise backgrounds. Thus, the average power spectrum of the images in Obj-MNIST is identical to the backgrounds of Obj-MNIST-PSbg. Using this method, the ensemble spectral content of foreground objects is matched to the scene in which they occur. Sample images from the variants can be seen in Figure 3.

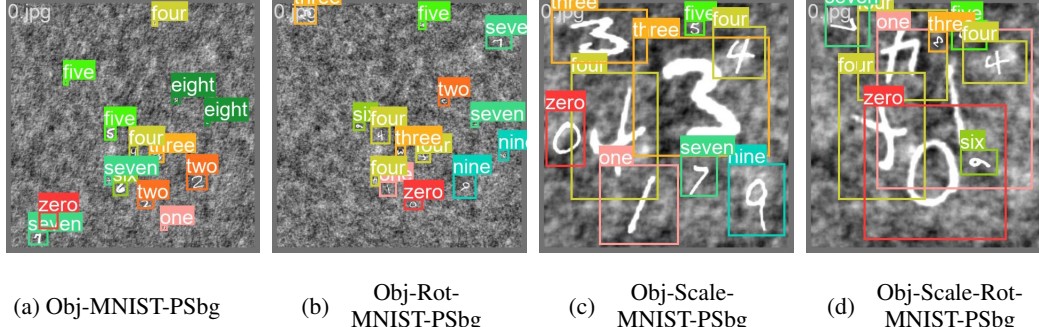

(a) Obj-MNIST-PSbg  (b) Obj-Rot-MNIST-PSbg  (c) Obj-Scale-MNIST-PSbg  (d) Obj-Scale-Rot-MNIST-PSbg

Figure 3: Scaled examples from the validation split of our proposed Obj-MNIST-PSbg datasets.

## 5 EXPERIMENTS

In order to evaluate invariance and each model's ability to generalize across applications, we test in two distinct environments, i) our proposed MOD datasets and -PSbg variants, and ii) Microsoft's COCO dataset (Lin et al., 2014). The COCO experiments gauge performance on a standard, generalized object detection task and are based on the standard training practices for YOLOv9 (Wang et al., 2024; Jocher et al., 2023).

### 5.1 MOD DATASETS EXPERIMENTS

The experiments on the MOD datasets and their -PSbg variants focus on gauging the degree of invariance in a model when rotation, scale, or rotation and scale are present in the dataset. To this end, Y9t and the three proposed models, $C_4$-Y9t, S-Y9t, and $C_4$S-Y9t, are trained on the Obj-MNIST dataset for 25 epochs. After each epoch, the models are validated on the Obj-Rot-MNIST, Obj-Scale-MNIST, and Obj-Scale-Rot-MNIST datasets. These validations have no impact on the training, and

---

[3]To be published alongside this work and included in the supplementary material.

consist entirely of unseen digits. The goal is to gauge how the degree of invariance for an unseen transformation group changes as training progresses on the standard non-transformed dataset. We use the same environment and methodology on the -PSbg datasets. Using the weights from the epoch with the highest mAP 50-95 on standard validation data, we test using unseen partitions. For each dataset variant, we create 30 subsets consisting of 992 images sampled with replacement from a completely unseen partition, averaging their results (Fisher transform). Model performance is reported as Table 3. Standard error was negligible and thus omitted. Bold and underlined scores denote the highest model performance for each MOD dataset and -PSbg variant. Performance results during training are given as Figure 4.

Table 3: Obj-MNIST and Obj-MNIST-PSbg experimental results

| | | Standard | | Rot | | Scale | | Scale-Rot | |
|---|---|---|---|---|---|---|---|---|---|
| | | mAP 50 | mAP 50-95 | mAP 50 | mAP 50-95 | mAP 50 | mAP 50-95 | mAP 50 | mAP 50-95 |
| **Obj-MNIST** | Y9t | 0.98425 | 0.77192 | 0.29948 | 0.09877 | 0.02717 | 0.01393 | 0.00571 | 0.00150 |
| | $C_4$-Y9t | 0.98660 | **0.79473** | **0.41037** | **0.15655** | 0.04775 | 0.02540 | 0.01068 | 0.00323 |
| | S-Y9t | 0.98694 | 0.75723 | 0.31100 | 0.10135 | **0.17966** | **0.09859** | 0.02890 | 0.00971 |
| | $C_4$S-9t | **0.98900** | 0.79332 | 0.40539 | 0.15132 | 0.15880 | 0.09594 | **0.04083** | **0.01536** |
| **Obj-MNIST-PSbg** | Y9t | 0.95243 | 0.67040 | 0.19753 | 0.05211 | 0.06431 | 0.03351 | 0.00737 | 0.00173 |
| | $C_4$-Y9t | 0.98607 | 0.72328 | **0.36630** | **0.11204** | 0.10142 | 0.05244 | 0.01218 | 0.00302 |
| | S-Y9t | 0.97577 | 0.69404 | 0.19578 | 0.05327 | 0.11306 | 0.05429 | 0.00974 | 0.00279 |
| | $C_4$S-Y9t | **0.98751** | **0.73134** | 0.28755 | 0.07938 | **0.11866** | **0.06311** | **0.01624** | **0.00437** |

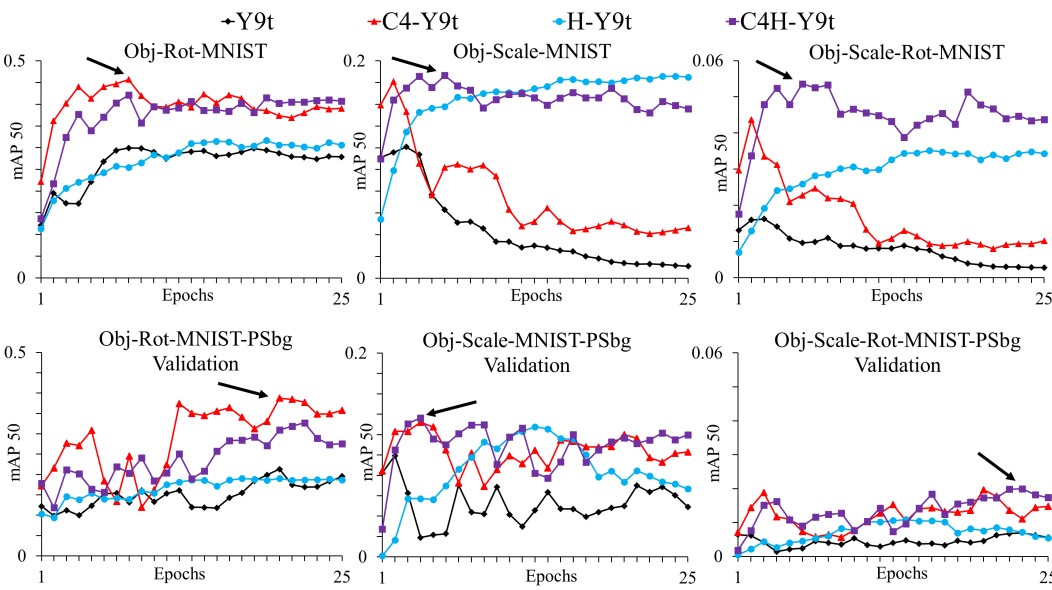

Figure 4: Validations taken on each proposed -PSbg dataset during training, with an arrow highlighting the top performance. The separate validation process had no impact on model training.

This experimental setup is similar to Maurel et al. (2023)'s, with the key difference of disallowing any amount of data augmentation. In their paper, data augmentation by random rotations of $\pm 90$ degrees and random mirroring on the x or y axis are applied to their generated datasets during training. By omitting these augmentations, we can explicitly attribute any performance differences between models on the validation sets strictly to the inclusion of the equivariant methods. Disallowing augmentations on the training data further ensures that no transformed data is seen by the models for the purpose of training, and that validating on the transformed datasets will be the model's first time encountering transformed data.

The results of the Obj-MNIST experiments are readily understood: By replacing convolutions with equivariant ones of relevant groups, the robustness to the transformation increases. We see an overall degradation of results in the experiments trained on the Obj-MNIST-PSbg datasets, which is expected. $C_4$S-Y9t appears to perform better overall on the -PSbg experiments for standard and scaled data, outperforming $C_4$-Y9t on standard data and S-Y9t on scaled data.

When analyzing the graphs, it is best to compare our proposed models individually to Y9t. This comparison explicitly demonstrates the improvement over the standard model as training progresses. The top row shows the results on the standard MOD datasets, where the benefit from equivariance is gained near the beginning of training. The bottom row shows that robustness to relevant transformations generally grows as training progresses in the -PSbg experiments.

## 5.2 COCO EXPERIMENTS

To quantify the effects of equivariant representations on object recognition in natural image datasets, we consider the COCO 2017 dataset (Lin et al., 2014) consisting of over 200,000 annotated images with 80 object categories. In particular, we train each proposed model ($C_4$-Y9t, S-Y9t, and $C_4$S-Y9t) on this dataset, as well as the base verion of Y9t as a model control, to determine how convolutions equivariant to rotation, scale, or both affect Y9t-family models on general object-detection tasks. To this end, all training hyperparameters of the Y9t-family models, including those for augmentations, are held constant(Wang et al., 2024; Jocher et al., 2023).

Table 4: COCO 2017 experimental results

| | Precision | Recall | mAP 50 | mAP 50-95 |
|---|---|---|---|---|
| Y9t | 0.61113 | 0.47207 | 0.51655 | 0.36996 |
| $C_4$-Y9t | **0.65772** | **0.50975** | **0.56443** | **0.41959** |
| S-Y9t | 0.61022 | 0.46505 | 0.50505 | 0.36975 |
| $C_4$S-Y9t | 0.62312 | 0.49192 | 0.53632 | 0.39847 |

$C_4$-Y9t and $C_4$S-Y9t show improvements over the standard Y9t. The most impressive of our proposed models is $C_4$-Y9t, which improves on all measured metrics by at least 3.7%. $C_4$-Y9t similarly improves on all metrics by at least 1.1%. These results show that the Y9t model is already quite effective at scale invariance when scales are varied during training. The additional scale invariance provided by S-Y9t is not sufficient to compensate for the reduction in the number of parameters. This result suggests that scale variance is the largest contributor to mAP loss in low-parameter object detection models.

The difference in performance between $C_4$S-Y9t and $C_4$-Y9t on COCO is likely for similar reasons to S-Y9t's reduction in quality. While there is indeed a high amount of existing scale invariance incorporated into the Y9t model, there is no explicit rotational invariance. The lack of existing rotational invariance paired with the simple yet effective method of cardinally rotating convolutional kernels from Cohen & Welling (2016) shows a clear improvement, likely due to introducing a high degree of invariance to a symmetry group unrepresented explicitly, namely, rotation.

## 6 CONCLUSION

We proposed the $C_4$-Y9t, S-Y9t, and $C_4$S-Y9t models to gauge the impact of various equivariant convolutions in the parameter-efficient Y9t model. Testing on our proposed MOD datasets, we found that each proposed model improves robustness to transformations pertaining to its supported symmetry group. Similar improvements were also seen on COCO 2017, a general object detection dataset, where both $C_4$-Y9t and $C_4$S-Y9t perform better than the Y9t baseline.

In future work, the running-time efficiency of equivariance methods could be improved. Rotational methods have greater time requirements than standard convolutions, while scale methods take even longer, as shown by Table 1. Following optimization, experimentation of equivariant methods on larger object detection models, such as YOLOv9-c would be valuable.

The MNIST Object Detection Datasets Generation Framework could also be improved to support more transformations and other background methods. The framework could be extended to support features beyond the MNIST digits, such as using segmentation labels as masks to extract the 80 classes of objects from the COCO 2017 dataset. Support could be added for transformed bounding boxes for more general equivariance support.

## 7 REPRODUCIBILITY STATEMENT

In order to allow for our results to be replicated elsewhere, we will publish our dataset generation tool alongside this work on GitHub[4]. This tool allows for the generation of all datasets proposed in this paper, along with other background configurations, such as pink noise. In the appendix we discuss the architecture of our proposed models, providing the precise implementation of each equivariance group. The models are constructed using the Ultralytics library of Jocher et al. (2023) as the foundation of our equivariant blocks and is used as our baseline Y9t implementation. We implement equivariant versions of each block using code published with Cohen & Welling (2016) and Worrall & Welling (2019).

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

# A  Appendix

In this appendix we first describe the architectures of our proposed models to show how equivariance is used throughout them. Then we provide additional background material regarding group theory and the definition of equivariance, largely based on an appendix in the novel by Weiler et al. (2025). For further reading on the motivations behind equivariance, we suggest the essays by Olah et al. (2020; 2021).

## A.1  Architecture of Proposed Models

We use the code provided by the Ultralytics library (Jocher et al., 2023) as a baseline to create our equivariant versions. We further use the rotation implementation by Cohen & Welling (2016) and the scale implementation by Worrall & Welling (2019). We implement two versions of every block in Y9t, one for each symmetry group, and combine them following the Y9t architecture as listed in the Ultralytics library. Below we include the construction details for each proposed model. In these tables we use "Cha" as shorthand for the number of channels, "k" to represent the kernel size, and "str" to represent the stride. Some blocks may require multiple channels or inputs. For the concat layers, all data is concatenated on the channel dimension. Finally, "sf" represents the scale factor for upsampling layers.

|  | Index | Previous Layer Index | Layer | Layer Parameters |
|---|---|---|---|---|
| Backbone | 0 | (Input) | Conv | Cha: 16, k:3, str: 2 |
|  | 1 | 0 | Conv | Cha: 32, k:3, str: 2 |
|  | 2 | 1 | ELAN1 | Cha: 32, 32, 16 |
|  | 3 | 2 | AConv | Cha: 64 |
|  | 4 | 3 | RepNCSPELAN4 | Cha 64, 64, 32, k: 3 |
|  | 5 | 4 | AConv | Cha: 96 |
|  | 6 | 5 | RepNCSPELAN4 | Cha: 96, 96, 48, k: 3 |
|  | 7 | 6 | AConv | Cha: 128 |
|  | 8 | 7 | RepNCSPELAN4 | Cha: 128, 128, 64, k: 3 |
|  | 9 | 8 | SPPELAN | Cha: 128, 64 |
| Neck | 10 | 9 | Upsample | sf: 2, Mode: Nearest |
|  | 11 | 6, 10 | Concat |  |
|  | 12 | 11 | RepNCSPELAN4 | Cha: 96, 96, 48, k: 3 |
|  | 13 | 12 | Upsample | sf: 2, Mode: Nearest |
|  | 14 | 4, 13 | Concat |  |
|  | 15 | 14 | RepNCSPELAN4 | Cha: 64, 64, 32, k: 3 |
|  | 16 | 15 | AConv | Cha: 48 |
|  | 17 | 12, 16 | Concat |  |
|  | 18 | 17 | RepNCSPELAN4 | Cha: 96, 96, 48, k: 3 |
|  | 19 | 18 | AConv | Cha: 64 |
|  | 20 | 9, 19 | Concat |  |
|  | 21 | 20 | RepNCSPELAN4 | Cha: 128, 128, 64, k: 3 |
|  | 22 | 15, 18, 21 | Detection Heads |  |

Table 5: Architecture of Y9t (Wang et al., 2024; Jocher et al., 2023).

|  | Index | Previous Layer Index | Layer | Layer Parameters |
|---|---|---|---|---|
| Backbone | 0 | (Input) | C4ConvZ2 | Cha: 16, k:3, str: 2 |
|  | 1 | 0 | C4ConvC4 | Cha: 16, k:3, str: 2 |
|  | 2 | 1 | C4ELAN1 | Cha: 16, 16, 8 |
|  | 3 | 2 | C4AConv | Cha: 32 |
|  | 4 | 3 | C4RepNCSPELAN4 | Cha 32, 32, 16, k: 3 |
|  | 5 | 4 | C4AConv | Cha: 48 |
|  | 6 | 5 | C4RepNCSPELAN4 | Cha: 48, 48, 24, k: 3 |
|  | 7 | 6 | C4AConv | Cha: 64 |
|  | 8 | 7 | C4RepNCSPELAN4 | Cha: 64, 64, 32, k: 3 |
|  | 9 | 8 | C4SPPELAN | Cha: 64, 32 |
| Neck | 10 | 9 | C4Upsample | sf: 2, Mode: Nearest |
|  | 11 | 6, 10 | Concat |  |
|  | 12 | 11 | C4RepNCSPELAN4 | Cha: 48, 48, 24, k: 3 |
|  | 13 | 12 | C4Upsample | sf: 2, Mode: Nearest |
|  | 14 | 4, 13 | Concat |  |
|  | 15 | 14 | C4RepNCSPELAN4 | Cha: 32, 32, 16, k: 3 |
|  | 16 | 15 | C4AConv | Cha: 24 |
|  | 17 | 12, 16 | Concat |  |
|  | 18 | 17 | C4RepNCSPELAN4 | Cha: 48, 48, 24, k: 3 |
|  | 19 | 18 | C4AConv | Cha: 32 |
|  | 20 | 9, 19 | Concat |  |
|  | 21 | 20 | C4RepNCSPELAN4 | Cha: 64, 64, 32, k: 3 |
|  | 22 | 15 | C4Flatten |  |
|  | 23 | 18 | C4Flatten |  |
|  | 24 | 21 | C4Flatten |  |
|  | 25 | 22, 23, 24 | Detection Heads |  |

Table 6: Architecture of our proposed $C_4$-Y9t.

| | Index | Previous Layer Index | Layer | Layer Parameters |
|---|---|---|---|---|
| Backbone | 0 | (Input) | SConvZ2 | Cha: 16, k:3, str: 2 |
| | 1 | 0 | SConvS | Cha: 27, k:3, str: 2 |
| | 2 | 1 | SELAN1 | Cha: 27, 27, 8 |
| | 3 | 2 | SAConv | Cha: 54 |
| | 4 | 3 | SRepNCSPELAN4 | Cha 54, 54, 27, k: 3 |
| | 5 | 4 | SAConv | Cha: 80 |
| | 6 | 5 | SRepNCSPELAN4 | Cha: 80, 80, 41, k: 3 |
| | 7 | 6 | SAConv | Cha: 109 |
| | 8 | 7 | SRepNCSPELAN4 | Cha: 109, 109, 54, k: 3 |
| | 9 | 8 | S-SPPELAN | Cha: 109, 54 |
| Neck | 10 | 9 | SUpsample | sf: 2, Mode: Nearest |
| | 11 | 6, 10 | Concat | |
| | 12 | 11 | SRepNCSPELAN4 | Cha: 80, 80, 41, k: 3 |
| | 13 | 12 | SUpsample | sf: 2, Mode: Nearest |
| | 14 | 4, 13 | Concat | |
| | 15 | 14 | SRepNCSPELAN4 | Cha: 54, 54, 27, k: 3 |
| | 16 | 15 | SAConv | Cha: 41 |
| | 17 | 12, 16 | Concat | |
| | 18 | 17 | SRepNCSPELAN4 | Cha: 80, 80, 41, k: 3 |
| | 19 | 18 | SAConv | Cha: 54 |
| | 20 | 9, 19 | Concat | |
| | 21 | 20 | SRepNCSPELAN4 | Cha: 109, 109, 54, k: 3 |
| | 22 | 15 | SFlatten | |
| | 23 | 18 | SFlatten | |
| | 24 | 21 | SFlatten | |
| | 25 | 22, 23, 24 | Detection Heads | |

Table 7: Architecture of our proposed S-Y9t.

| | Index | Previous Layer Index | Layer | Layer Parameters |
|---|---|---|---|---|
| Backbone | 0 | (Input) | C4ConvZ2 | Cha: 16, k:3, str: 2 |
| | 1 | 0 | C4ConvC4 | Cha: 16, k:3, str: 2 |
| | 2 | 1 | C4ELAN1 | Cha: 16, 16, 16 |
| | 3 | 2 | C4AConv | Cha: 32 |
| | 4 | 3 | C4RepNCSPELAN4 | Cha 32, 32, 16, k: 3 |
| | 5 | 4 | C4AConv | Cha: 48 |
| | 6 | 5 | C4RepNCSPELAN4 | Cha: 48, 48, 24, k: 3 |
| | 7 | 6 | C4AConv | Cha: 64 |
| | 8 | 7 | C4RepNCSPELAN4 | Cha: 64, 64, 32, k: 3 |
| | 9 | 8 | C4SPPELAN | Cha: 64, 32 |
| | 10 | 9 | C4Flatten | |
| Neck | 11 | 10 | Upsample | sf: 2, Mode: Nearest |
| | 12 | 6 | C4Flatten | |
| | 13 | 11, 12 | Concat | |
| | 14 | 13 | SConvZ2 | Cha: 72, k: 3 |
| | 15 | 14 | SRepNCSPELAN4 | Cha: 72, 72, 35, k: 3 |
| | 16 | 15 | SFlatten | |
| | 17 | 16 | Upsample | sf: 2, Mode: Nearest |
| | 18 | 4 | C4Flatten | |
| | 19 | 17, 18 | Concat | |
| | 20 | 19 | SConvZ2 | Cha: 47, k: 3 |
| | 21 | 20 | SRepNCSPELAN4 | Cha: 47, 47, 24, k: 3 |
| | 22 | 21 | SFlatten | |
| | 23 | 22 | AConv | Cha: 48 |
| | 24 | 16, 23 | Concat | |
| | 25 | 24 | SConvZ2 | Cha: 72, k: 3 |
| | 26 | 25 | SRepNCSPELAN4 | Cha: 72, 72, 35, k: 3 |
| | 27 | 26 | SFlatten | |
| | 28 | 27 | AConv | Cha: 64 |
| | 29 | 10, 28 | Concat | |
| | 30 | 29 | SConvZ2 | Cha: 96, k: 3 |
| | 31 | 30 | SRepNCSPELAN4 | Cha: 96, 96, 47, k: 3 |
| | 32 | 31 | SFlatten | |
| | 25 | 22, 27, 32 | Detection Heads | |

Table 8: Architecture of our proposed $C_4$S-Y9t.

## A.2 INVARIANCE AND EQUIVARIANCE

Invariance is defined differently in a mathematical context than when used in the context of machine learning. In mathematics, an invariant is defined as follows:

> *"A property, quantity, or relationship that is not changed by collection of specific operations or transformations, or by how that property is observed is called an invariant. For example, the distance between two points in a plane does not change under a rotation. Thus, distance is invariant of rotations"* Tanton (2005).

Invariance can then be defined as the property of being invariant. When discussing a neural network having invariance, the term is used somewhat loosely. Typically a neural network is neither definitively invariant nor definitively not invariant to a transformation. Instead, it is described as having a degree of invariance to the transformation. For example, if a neural network has its activations slightly altered by a transformation but produces nearly the same output as it did for the untransformed input, it is said to be highly invariant to that transformation. The neural network may also be described as having a high degree of robustness to this transformation. In much of the literature in this field, the term "degree of invariance" is replaced by "invariance", and when a neural network is called "invariant" to a transformation, typically the intended meaning is that it has a high degree of invariance to the transformation. This difference from the mathematical definition is due to the nature of a neural network, which aims to approximate a target function rather than represent it exactly. Figure 5a shows this property well: while activations may subtly differ in a neural network when analyzing the reflected cat image rather than the original image, the resulting classification is unchanged, and thus the neural network is considered invariant to reflections (Weiler et al., 2025).

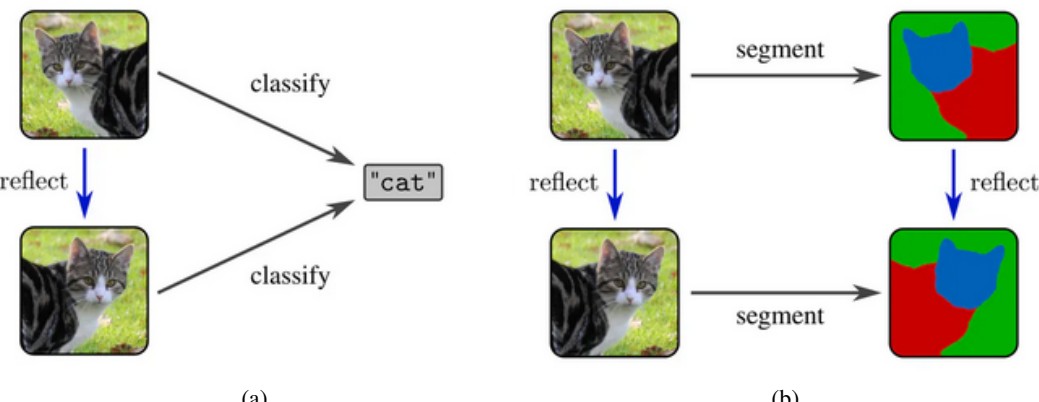

(a) (b)

Figure 5: Directly adapted from Weiler et al. (2025). (a) An example of invariance on reflection: regardless of the reflection on the input image, the output is invariant and the image is correctly classified. (b) An example of equivariance on reflection: regardless of whether the reflection is applied before or after the segmentation, the output is the same.

In order to explain equivariance, we will briefly discuss the basics of group theory. A *group* consists of a set $G$ and a binary operator $\star$ on $G$ satisfying the following conditions (Weiler et al., 2025):

1. There exists an element $e \in G$ such that for any $g \in G$ one has $e \star g = g = g \star e$.

2. For any $g \in G$ there exists a $g^{-1} \in G$ such that $g \star g^{-1} = e = g^{-1} \star g$

3. For all $g, h, k \in G$, $(g \star h) \star k = g \star (h \star k)$

A simple example of a group is $(\mathbb{Z}, +)$, because any two integers may be added to find another integer. It is clear that the three properties hold. In most cases, we are concerned about equivariance using a *symmetry group*. The elements of the set G for a symmetry group are transformations pertaining to a certain symmetry and the binary operator combines these elements through composition or matrix multiplication.

Formally, equivariance is defined with respect to group actions on functions. The following explanation for left and right group actions, as well as equivariant maps are provided by Weiler et al.'s book on the subject of equivariance in convolutional neural networks (Weiler et al., 2025).

**Definition 1 (Left Group Action, Weiler et al. (2025))** *Let $G$ be a group and $X$ be a set. A left group action is a map*

$$\triangleright : G \times X \to X, \quad (g, x) \mapsto g \triangleright x$$

*that is compatible with the group composition and identity element:*

$$\begin{aligned} \text{associativity:} \quad & (gh) \triangleright x = g \triangleright (h \triangleright x) \text{ for any } g, h \in G, x \in X \\ \text{identity:} \quad & e \triangleright x = x \text{ for any } x \in X \end{aligned}$$

*$G$ is then said to act on X from the left and X said to be a left G-set.*

A concrete example of the left group action loosely based on one in Weiler et al. (2025) is as follows: Let $x \in \mathbb{R}^2$ and $G = SO(2)$, where $SO(2)$ is the special orthogonal group in 2D, which can be thought of as a $2 \times 2$ matrix representing rotations on a plane. We further have $g, h \in G$, with $g$ representing a rotation by $30°$ clockwise about the origin, and $h$ a rotation by $15°$ in the same direction. The $(gh) \triangleright x$ term indicates that the two rotations are composed to create a single group element of $G$ corresponding to a $45°$ rotation and then used to act on x. We may rotate $x$ with $g$, where $g \triangleright x$ now represents the point rotated $30°$ clockwise about the origin, with $g$ acting on $x$ via matrix multiplication with $g \times x$. Similarly, $g \triangleright (h \triangleright x)$ represents a point first rotated by $15°$ clockwise, then $30°$ clockwise. It is now clear that $(gh) \triangleright x = g \triangleright (h \triangleright x)$, or in other words, regardless of whether $g$ or $h$ are combined or computed on $x$ separately, the total rotation is $45°$. Similarly, the identity element of the $SO(2)$ group is $e \in SO(2)$ where $e$ represents the clockwise rotation by $0°$.

**Definition 2 (Right Group Action, Weiler et al. (2025))** *Let $G$ be a group and $X$ be a set. A map*

$$\triangleleft : X \times G \to X, \quad (x, g) \mapsto g \triangleleft x$$

*is denoted as [a] right group action iff it satisfies:*

$$\begin{aligned} \text{associativity:} \quad & x \triangleleft (gh) = (x \triangleleft g) \triangleleft h \text{ for any } g, h \in G, x \in X \\ \text{identity:} \quad & x \triangleleft e = x \text{ for any } x \in X \end{aligned}$$

$G$ is then said to act on $X$ from the right and $X$ is then said to be a right G-set. The difference between the two definitions is in their associativity formulae, and particularly in the ordering of $gh$ compositions. These two group actions may be freely converted into one another by inversion of the acting group elements because this swaps the order of the compositions. An equivariant map may be defined by either left or right group actions, but for brevity we only provide the definition for the left group action (Weiler et al., 2025).

**Definition 3 (Equivariant Map, Weiler et al. (2025))** *Let X and Y be G-sets, acted on by group action $\triangleright_X$ and $\triangleright_Y$, respectively. If a function $L : X \to Y$ commutes with these group actions,*

$$L(g \triangleright_X x) = g \triangleright_Y L(x) \quad \forall g \in G, x \in X,$$

*it is said to be G-equivariant. This condition corresponds to the commutative diagram below:*

$$\begin{array}{ccc} X & \xrightarrow{L} & Y \\ {\scriptstyle g \triangleright_X} \downarrow & & \downarrow {\scriptstyle g \triangleright_Y} \\ X & \xrightarrow{L} & Y \end{array}$$

Let us consider how the above commutative diagram applies to a convolution. Assume $X$ is the input, $Y$ is the output, and $L$ is a convolution on the data. The convolution is said to be equivariant to group or semigroup $G$, if for all transformations $g \in G$, the diagram holds true. Using this diagram, we can see that mathematically invariant functions are a special case of G-equivariant functions, where the output group action $g \triangleright_y = \text{id}_Y$ is trivial (Weiler et al., 2025):

$$
\begin{array}{ccc}
X & & \\
\big\downarrow g \triangleright x \quad \searrow^{L} & & \\
& Y \;\circlearrowleft\, \mathrm{id}_Y & \qquad \Longleftrightarrow \qquad \\
X & \nearrow_{L} &
\end{array}
\qquad
\begin{array}{ccc}
X & \xrightarrow{\;L\;} & Y \\
\big\downarrow g \triangleright x & & \big\downarrow \mathrm{id}_Y \\
X & \xrightarrow{\;L\;} & Y
\end{array}
\qquad \text{Weiler et al. (2025)}
$$

Here $\mathrm{id}_Y$ is the identity element for G-set Y. The term "equivariance," like "invariance," is used loosely in the field of machine learning rather than according to its formal mathematical definition. The contrast in the definition when discussing equivariance in machine learning is the same as when discussing invariance in machine learning: a neural network is said to be equivariant if it approximates true equivariance well. This equivariance may apply to an entire neural network, a portion of a network such as a layer or convolution, or even a single operation.

