# OpenReview forum: "Parameter-Efficient Invariance via Equivariant Object Detection Networks"
_ICLR.cc/2026/Conference — ICLR 2026 Conference Withdrawn Submission_

### Official Review · Reviewer_DVHD · 2025-10-25

**Soundness:** 3
**Presentation:** 2
**Contribution:** 2
**Rating:** 4
**Confidence:** 3

**Summary:**

- This paper proposes a method that leverages the rotational and scale symmetries present in image data to achieve parameter-efficient invariance. Specifically, the research focuses on the rotation group (C4 group) and scale, implementing improvements to the network. Experiments utilize the small-scale MNIST dataset for model refinement and evaluation. By incorporating an equivariance method into the Y9t network while maintaining a comparable number of learnable parameters, the paper discusses its effectiveness on unobserved rotations, scales, and rotation-scale-transformed partitions of the MOD dataset. Furthermore, these models are evaluated on MS COCO.

**Strengths:**

- Rotation and scale invariance in object detection is an important research topic.
- This paper is well-structured and easy to understand.
- Source code is shared as supplementary material.
- The attempt to incorporate the concept of rotation groups (more generally, group theory) into object detection modules is interesting.

**Weaknesses:**

- The primary weakness is its specialization in character recognition (character detection) based on MNIST.
  - The paper appears focused on characters, including the MOD dataset. As noted in related work, rotation invariance is crucial for satellite and aerial imagery. Why wasn't evaluation performed on datasets specialized for such imagery (e.g., DOTA or DOTA2.0)?
  - Furthermore, as seen in these studies, research targeting general objects (i.e., MSCOCO) has also been proposed.
  - As seen in these experiments, evaluation on rotated MSCOCO datasets would also be meaningful.
    - Kalra, Agastya, et al. “Towards rotation invariance in object detection.” Proceedings of the IEEE/CVF international conference on computer vision. 2021.
    - Shibata, Takashi, Masayuki Tanaka, and Masatoshi Okutomi. “Robustizing Object Detection Networks Using Augmented Feature Pooling.” Proceedings of the Asian Conference on Computer Vision. 2022.

- Insufficient experimentation regarding evaluation on MSCOCO
  - To my understanding, the evaluation metrics in the experiments (Table 3) differ from the publicly available values. Additional explanation is needed on this point. Furthermore, clarification is required on whether val or test-dev was used.
  - https://docs.ultralytics.com/ja/models/yolov9/

- Furthermore, why did performance improve compared to YOLOv9-t (Y9t)? More detailed discussion, experiments, and analysis are needed on this point. Are there samples within MSCOCO that require rotation-invariant concepts? What is their proportion?
  - More sophisticated models like YOLO11 have also been released. Is the effect also present in these models?

**Questions:**

Please refer to the weaknesses section for the thought process leading to these questions.

- This paper appears to focus specifically on text, including the MOD dataset. As mentioned in related research, rotation invariance is crucial for satellite images and aerial photos. Why was evaluation not performed on image datasets specialized for satellite/aerial photos (e.g., DOTA or DOTA2.0)?
- Why do the evaluation metrics in the experiments (Table 3) differ from the publicly available values?
- Did the performance improve compared to YOLOv9-t (Y9t) on the dataset using NSCOCO? (Discussion based on experimental facts is preferred)
- More sophisticated models than YOLO, such as YOLO11, have also been released. Are these models also effective?

---

### Official Review · Reviewer_j3MV · 2025-10-27

**Soundness:** 2
**Presentation:** 2
**Contribution:** 1
**Rating:** 2
**Confidence:** 4

**Summary:**

This paper proposes variants of the Y9t architecture with rotation and scale equivariant convolutions. A MNIST-based dataset for evaluate equivariance detection is proposed, and experimental results are presented between the baseline Y9t and the multiple equivariance-equipped variants, showing improved performance.

**Strengths:**

- The presentation of the paper is generally clear, with a well-defined motivation and outline.
- The proposed MNIST MOD dataset with power spectrum background is an interesting construction.

**Weaknesses:**

- The proposed architecture applies commonly known equivariant convolutions to Y9t, and lacks novelty or technical depth.
- The proposed MNIST MOD dataset also does not present sufficient novelty and contribution to the field.
- In the experiments, comparisons were only made against the baseline non-equivariant model Y9t, but not against any other equivariant models such as EQ-YOLO or ReDet as noted in related works. Also simple approaches to equivariant such as data augmentation should be compared with as well.
- As in Table 1, the inference time of the equivariant networks are over 10x higher. This limits the practical applicability of the work.
- As in Table 3, none of the proposed models actually perform well on the Scale-Rot datasets, despite being marginally better than the baseline. This raises serious concern on the actual performance of the proposed method.
- MNIST is ultimately a simple and non-realistic dataset, compared with images in the wild.

**Questions:**

- How does the proposed architecture compare with other equivarient architectures in the literature, and with data augmentation?
- How to address the major increase in computational cost in the proposed model?

---

### Official Review · Reviewer_yDQ9 · 2025-10-28

**Soundness:** 2
**Presentation:** 1
**Contribution:** 2
**Rating:** 2
**Confidence:** 4

**Summary:**

This paper introduces parameter-efficient geometric invariance to object detectors by incorporating equivariant convolutions ($C_4$ for rotations and $S$ for scales) into the YOLOv9-t (Y9t) architecture, yielding three variants ($C_4$-Y9t, S-Y9t, $C_4$S-Y9t) with parameter counts comparable to the baseline. For evaluation, this paper presents the MNIST Object-Detection (MOD) datasets, which systematically manipulate digit rotations and scales, including a Power Spectrum background variant (-PSbg) for added realism. Experiments reveal substantial mAP gains on unseen transformations in MOD without augmentation. The models also transfer effectively to COCO 2017, where $C_4$-Y9t improves mAP by +4.9% over the baseline.

**Strengths:**

1. The paper tackles the well-defined and important challenge of achieving parameter-efficient geometric invariance in object detectors. This is a highly relevant goal for deploying models on resource-constrained devices.

2 .This paper proposes a new MNIST Object-Detection dataset, which provides a clean, controlled environment to quantitatively evaluate a model's robustness to specific, isolated transformations (rotation, scale).

**Weaknesses:**

1. The core equivariant operations ($C_4$ and $S$) lack novelty, citing foundational work from Cohen & Welling (2016) and Worrall & Welling (2019), respectively. The key contribution is not enough.

2. While the paper focuses on parameter-efficiency, it largely overlooks computational-efficiency. Table 1 shows that the best model, $C_{4}$-Y9t, is slower than Y9t (5.5 ms vs 2.4 ms). More alarmingly, the scale-equivariant S-Y9t is 14.2x slower than Y9t (34.0 ms vs 2.4 ms). This severe speed degradation is a major limitation that compromises the practical utility of the proposed methods, especially for YOLO-family models prized for their speed.

3. The architecture of the combined $C_{4}S$-Y9t model appears overly complex and lacks a supporting figure for clarity.

4. The paper uses the $C_{4}$ group (cardinal rotations, i.e., 90 degrees) for its rotation-equivariant model. However, its own Obj-Rot-MNIST dataset tests on arbitrary rotations in $[0, 359]^{\circ}$. While $C_{4}$ clearly provides a strong benefit, it's a very coarse approximation of the problem. The paper mentions $SO(2)$ (continuous rotations) but doesn't justify the choice of $C_{4}$ beyond preliminary testing.

**Questions:**

1. Please explain the reason for the 14.2x slowdown of S-Y9t. Since speed is a defining characteristic of YOLO, how can we understand this trade-off?

2. Please provide more insight into the $C_{4}S$-Y9t architecture. Why is there a need to project $C_{4}$ features back to $\mathbb{R}^{2}$ before the neck, only to immediately lift them to $S$? What about simpler alternatives, such as maintaining the $C_{4}$ representation throughout the neck or using a unified $C_{4}S$ group?

3. Why was the $C_{4}$ group selected instead of a finer group like $C_{8}$ or the continuous $SO(2)$ group, especially considering that the MOD-Rot dataset evaluates arbitrary rotations?

---

### Official Review · Reviewer_ge6y · 2025-11-10

**Soundness:** 1
**Presentation:** 1
**Contribution:** 1
**Rating:** 2
**Confidence:** 5

**Summary:**

This paper integrates rotational and scale equivariant convolutions into a compact object detector (YOLOv9-t), yielding three variants: C4-Y9t (rotation, p4/C4), S-Y9t (scale, dilation-based semigroup S), and C4S-Y9t (rotation in backbone, scale in neck). The authors introduce new MNIST Object-Detection (MOD) datasets, to directly evaluate rotation- and scale-invariant methods in object detection networks, and allow researchers concretely and consistently gauge changes to robustness in the face of transformations. They additionally evaluate on COCO 2017. With approximately matched parameter counts to YOLOv9-t, they report mAP improvements on unseen rotation, scale transportations (+16.8%, +15.2%, 3%).

**Strengths:**

1. The problem and solution are very clear,  integrateing rotational and scale equivariant convolutions into a compact object detector (YOLOv9-t), and yielding three variants: C4-Y9t (rotation, p4/C4), S-Y9t (scale, dilation-based semigroup S), and C4S-Y9t (rotation in backbone, scale in neck).
2. A modified MNIST datasets is introduced to evaluate objects with different scale and rotation.
3. The proposed methods' performance are better than the baseline YOLOv9-t.

**Weaknesses:**

1. Overall, the paper lacks of novelty.
2. The writing of this manuscript has a big space to improve.
3. The contribution of method is limited, and not well presented.
4. The baseline is chosen with only Yolov9-t.

**Questions:**

1. The authors should give a more clear introduction about the mehod and dataset improvement.
2. There are only datasets with different size and angle objects, such as arieal images, cars, boths, the authors should evaluate their methods on these datasets.
3. More related methods should be compared.

---

### Note · Authors · 2025-11-12

**Comment:**

We would like to thank the reviewers for their thorough analysis.  We will revise our manuscript to better reflect our work for future submission.

**Withdrawal Confirmation:**

I have read and agree with the venue's withdrawal policy on behalf of myself and my co-authors.